# Liquid–Liquid Phase Separation Enhances TDP-43 LCD Aggregation but Delays Seeded Aggregation

**DOI:** 10.3390/biom11040548

**Published:** 2021-04-08

**Authors:** Donya Pakravan, Emiel Michiels, Anna Bratek-Skicki, Mathias De Decker, Joris Van Lindt, David Alsteens, Sylvie Derclaye, Philip Van Damme, Joost Schymkowitz, Frederic Rousseau, Peter Tompa, Ludo Van Den Bosch

**Affiliations:** 1Department of Neurosciences, Experimental Neurology and Leuven Brain Institute (LBI), KU Leuven-University of Leuven, 3000 Leuven, Belgium; donya.pakravan@kuleuven.vib.be (D.P.); mathias.dedecker@kuleuven.be (M.D.D.); philip.vandamme@uzleuven.be (P.V.D.); 2VIB, Center for Brain & Disease Research, Laboratory of Neurobiology, 3000 Leuven, Belgium; 3Switch Laboratory, VIB-KU Leuven Center for Brain & Disease Research, 3000 Leuven, Belgium; emiel.michiels@kuleuven.be (E.M.); joost.schymkowitz@kuleuven.be (J.S.); frederic.rousseau@kuleuven.be (F.R.); 4VIB-VUB Center for Structural Biology, 1050 Brussels, Belgium; anna.bratek-skicki@vub.be (A.B.-S.); joris.van.lindt@vub.be (J.V.L.); peter.tompa@vub.be (P.T.); 5Institute of Life Sciences, Université Catholique de Louvain, 1348 Louvain-la-Neuve, Belgium; david.alsteens@uclouvain.be (D.A.); sylvie.derclaye@uclouvain.be (S.D.); 6Department of Neurology, University Hospitals Leuven, 3000 Leuven, Belgium

**Keywords:** phase separation, aggregation, amyotrophic lateral sclerosis, TDP-43

## Abstract

Aggregates of TAR DNA-binding protein (TDP-43) are a hallmark of several neurodegenerative disorders, including amyotrophic lateral sclerosis (ALS). Although TDP-43 aggregates are an undisputed pathological species at the end stage of these diseases, the molecular changes underlying the initiation of aggregation are not fully understood. The aim of this study was to investigate how phase separation affects self-aggregation and aggregation seeded by pre-formed aggregates of either the low-complexity domain (LCD) or its short aggregation-promoting regions (APRs). By systematically varying the physicochemical conditions, we observed that liquid–liquid phase separation (LLPS) promotes spontaneous aggregation. However, we noticed less efficient seeded aggregation in phase separating conditions. By analyzing a broad range of conditions using the Hofmeister series of buffers, we confirmed that stabilizing hydrophobic interactions prevail over destabilizing electrostatic forces. RNA affected the cooperativity between LLPS and aggregation in a “reentrant” fashion, having the strongest positive effect at intermediate concentrations. Altogether, we conclude that conditions which favor LLPS enhance the subsequent aggregation of the TDP-43 LCD with complex dependence, but also negatively affect seeding kinetics.

## 1. Introduction

Transactivation response element (TAR) DNA-binding protein 43 (TDP-43) is a DNA/RNA-binding protein implicated in transcriptional and post-transcriptional regulation of gene expression. Under normal conditions, the protein is primarily located in the nucleus, while it also shuttles between the nucleus and the cytoplasm. TDP-43 regulates RNA metabolism, RNA transport and RNA splicing [1,2]. The link between TDP-43 and neurodegenerative diseases was established when TDP-43 was found to be the major component of pathological ubiquitin-positive protein inclusions in patients with amyotrophic lateral sclerosis (ALS) and frontotemporal lobar degeneration (FTLD) [3,4]. Moreover, heterozygous missense mutations in the TAR DNA-binding protein (TARDBP) gene encoding TDP-43 are a genetic cause of ALS [5,6]. Since then, TDP-43 aggregates have also been observed in other neurodegenerative diseases [7,8,9]. Besides TDP-43, other RNA-binding proteins, such as FUS, also form aggregates, and this has greatly contributed to the general concept that protein misfolding and aggregation are mechanistically linked with a broad range of neurodegenerative diseases [10]. As a consequence, understanding the molecular events underlying protein misfolding and aggregation not only offers a better understanding of these diseases, but it could also open novel avenues for the development of drugs to interfere with their initiation and/or progression.

Recently, many ALS-related, aggregation-prone proteins have been shown to undergo liquid–liquid phase separation (LLPS), a process where biomolecules de-mix from a homogeneous solution and form liquid-like, high-density droplets within the low-density protein solution, potentially leading to the formation of aggregates as summarized in the following reviews [11,12]. LLPS also has many important physiological functions, being the main driving force for the generation of membraneless organelles (MLOs), such as stress granules (SGs) and P-bodies in the cytoplasm and nucleoli in the nucleus (reviewed in [13,14]). The function of SGs, which consist of RNA and RNA-binding proteins, is to dynamically regulate RNA storage, metabolism and translation during cellular stress [15,16]. Under pathological conditions, elicited by disease-causing mutations or sustained stress, SG homeostasis is disrupted and disturbed organelles are thought to lead to the production of pathological aggregates [17,18,19]. This model is also supported by observations that chronic optogenetic induction of stress granules is cytotoxic and shows mechanistic parallels with the evolution of ALS- frontotemporal dementia (FTD) pathology [20].

TDP-43 contains two tandem RNA-binding domains (RRMs) and a low-complexity domain (LCD) at the C-terminal end (TDP-43 LCD). LCDs are characterized by a low diversity of amino acids biased towards polar and charged ones, although the LCD of TDP-43 is not rich in charged residues [11]. The TDP-43 LCD is an intrinsically disordered domain, i.e., it lacks a well-defined structure under native conditions. Because of its multivalent nature, this domain contributes greatly to the LLPS of TDP-43, by utilizing the forces and balance between attractive hydrophobic and repulsive electrostatic interactions [21,22]. Of basic relevance to the mechanism(s) of disease, almost all ALS-related mutations in TDP-43 are localized within the LCD [23]. These mutations perturb LLPS by disrupting physiological protein–protein interactions and by affecting the stabilization of a transient helix within the LCD [21,24]. The LCD can be further divided into a glycine-rich region, a strong hydrophobic region, a glutamine/asparagine-rich region and a second glycine-rich region. The glutamine/asparagine-rich region resembles physiological yeast prion proteins, such as Sup35 or Mot3, and as a consequence it is also termed the prion-like domain (PLD). This domain has the inherent capacity to switch from an intrinsically disordered state to a potentially harmful, self-templating fibrillar state [25,26]. 

Recent results showed that the formation of liquid droplets by LLPS is often followed by their “maturation”, when they undergo a gradual transition to a less dynamic, gel-like state followed by solidification, such as aggregation [27,28,29]. For many disease-associated proteins, a direct link between LLPS and aggregation has been established, and for TDP-43 LCD, it has also been shown that certain LLPS-promoting conditions favor LCD fibrillation [21,30]. The exact mechanistic link between droplet formation and aggregation has not been established. Even the question of whether LLPS is mandatory for aggregation has not been unequivocally answered. Two additional mechanistic elements further complicate this link. First, the formation of aggregates is seeded, i.e., preformed aggregates (or their fragments, i.e., “seeds”) promote faster aggregation even in the absence of LLPS. Second, the speed with which this aggregation event occurs depends on protein–protein interactions, which are influenced by numerous factors, embodied in the type of seeds. TDP-43 is an RNA-binding protein, and RNA appears to have an effect on LLPS in general [31] and that of TDP-43 LCD in particular [32], which may further influence subsequent aggregation. Despite TDP-43 having cytosolic roles outside the context of SGs and without being toxic, it has been shown that when surrounded by RNA, it is more likely to undergo de-mixing leading to disease phenotypes [33,34]. If the protein cannot bind RNA, which is more likely to happen in the cytoplasm where the RNA concentration is much lower than in the nucleus, its tendency to form aggregates is enhanced, as was shown by optogenetic modulation [33]. In the nucleus, LLPS of TDP-43 is largely suppressed by the presence of RNA at high concentrations [31].

Whereas the connection between LLPS and aggregation, and the effect of RNA thereof, have been assessed under some conditions (e.g., pH, salt, urea), and the two processes are apparently mechanistically connected, their exact mechanistic link is difficult to dissect, because they rely on different elementary steps and ionic strength, for example, may affect them differently. As the interplay between LLPS and seeding aggregation of TDP-43 LCD had not been studied in detail, our aim was the detailed characterization of this relationship via a broad buffer screen to find a range of distinct conditions and thus uncouple LLPS from aggregation. To investigate seeding, we chose peptides corresponding to aggregation-prone regions (APRs) of LCD and applied them under LLPS promoting (LLPS+) and inhibiting (LLPS−) conditions. We observed a complex effect of phase separation on both the maturation of these dynamic condensates and their seeded “prion-like” aggregation. These results highlight a previously unrecognized role of LLPS in aggregation by affecting its aggregation kinetics. 

## 2. Materials and Methods

### 2.1. Cloning, Expression and Purification of TDP-43

The human TDP-43 DNA sequence (UniProtKB-Q13148 (TADBP_HUMAN)) was codon-optimized for bacterial expression, preceded by a 6X histidine tag, and located in a prokaryotic expression vector plasmid (CTD TDP-43) with a kanamycin resistance gene as a selection marker, preceded by lac operon, and was a kind gift from Prof. N. Fawzi (Department of Molecular Pharmacology, Physiology, and Biotechnology, Brown University, Providence, RI, USA) (Addgene plasmid # 98,669; http://n2t.net/addgene:98669; RRID:Addgene_98669) [21]. *Escherichia coli* BL21 STAR were transformed with the TDP-43 containing plasmid, and grown overnight. Cells were lysed by sonication on ice in 500 mM NaCl, 2 mM Tris HCl, 1 mM 1,4-dithiothreitol (DTT), protease inhibitors (cOmplete, EDTA-free Protease Inhibitor Tablets) (Roche Applied Science, Penzberg, Germany), 0.5 mM benzamidine hydrochloride (Sigma-Aldrich, Saint Louis, MO, USA) and 0.1 mM of phenylmethanesulfonyl fluoride (PMSF) solution (Sigma-Aldrich). The pellet was resolubilized in a denaturing buffer: 20 mM Tris HCl, 8 M urea, 500 mM NaCl, 10 mM imidazole, 1 mM DTT, pH 8.0 and sonicated in ice for 30 min. The solution was centrifuged at 20,000× *g* for 1 h at 4 °C. The supernatant was filtered through a filter of 0.45 μm and loaded onto a nickel-charged immobilized metal chelate affinity chromatography (IMAC) column HistrapTM HP (Cytiva (formally known as GE Healthcare), Buckinghamshire, UK), to which the 6x HIS tag of TDP-43 LCD binds. Bacterial debris was washed away with a denaturing buffer (8 M urea, 20 mM Tris buffer, 500 mM NaCl, 10 mM imidazole, 1 mM DTT, pH 8.0), after which the bound TDP-43 LCD was eluted with a linear 0 mM to 500 mM imidazole (Merck, Billerica, MA, USA) gradient. 

The protein fractions were combined, and the protein was buffer-exchanged into 2-(N-morpholino) ethanesulfonic acid (MES) buffer 20 mM, 3 M urea, pH 7.0. The 6xHis-tag domain was cleaved by incubating the samples with the tobacco etch virus protease (TEV, NEB, Ipswich, MA, USA) (1:10 protein/TEV molar ratio). All samples were incubated overnight at 34 °C without shaking. The TDP-43 LCD was separated from TEV using a nickel-charged IMAC column (Chicago, IL, USA) with 20 mM tris HCl, 8 M urea, 500 mM NaCl, 10 mM imidazole, 1 mM DTT, pH 8.0. TEV was eluted with a linear 0 mM to 500 mM imidazole gradient. With gel filtration using 8 M urea, 20 mM MES, pH 7.0 buffer, salts and other impurities were removed through the resin beads. After gel filtration, the protein was concentrated (Vivaspin^®^ 20, 3000 MWCO PES filter, Sartorius, Göttingen, Germany) and buffer-exchanged into MES buffer 20 mM, pH 5.5 to keep the protein soluble, without undergoing LLPS. Proteins were filtered through a 0.22 μm filter. Protein concentration was determined by Qubit (Life Technologies, Eugene, OR, USA) and absorption at 280 nm. The effectiveness of each purification step was determined by gel electrophoresis with Page Blue staining (Thermo Scientific, Rockford, IL, USA) to visualize the protein band. Pure protein samples were flash frozen and kept at −80 °C at a concentration of 0.7 mg/mL.

### 2.2. Peptide Synthesis

Peptides were generated by Genescript (The Netherlands) to total concentrations of 2–3 mg/mL per peptide. Hexafluoro isopropanol (HFIB) (Sigma-Aldrich) was added to lyophilized peptide so that samples could be maintained in coated film only to be redissolved when needed. Where specified, peptides were sonicated (Branson Digital sonifier 50/60 HZ) with 1-min cycles (15 pulses at 10% power with 30 s pauses), until completing a 1 min sonication time.

### 2.3. Thioflavin Kinetic Assays

To test aggregation of TDP-43 LCD, we used the amyloid-detection dye Thioflavin T (ThT) (Sigma-Aldrich, Saint Louis, MO, USA) in a Corning Black 96-well plate (Fisher, Kennebunk, ME, USA; catalogue N° 07-200-762). ThT is a fluorescent probe which is widely used to detect amyloid fibril formation. Upon binding to amyloid fibrils, a fluorescence signal is observed, which is indicative of cross-β-sheet formation which is due to the loss of internal aromatic ring rotation upon binding. However, ThT binding is not restricted to fibrillar cross-β-sheets, it also binds to some extent to oligomers and hydrophobic pockets via π–π interactions of aromatic residues, with low intensity [35]. ThT however does not change “kinetics”. Aggregation kinetics were obtained by combining 100 μL of the protein solution with a final concentration of 25 μM ThT. Fluorescence emission was monitored at 480-10 nm after excitation at 440-10 nm for ThT. ThS aggregation was measured using the Polar Star Omega (BMG Offenburg, Germany) laboratories at wavelength 430 nm. ThioS (Sigma-Aldrich Saint Louis, MO, USA) was used in cases where RNA was used. Where specified 1,6 hexanediol (Sigma-Aldrich, Dorset, UK) was spiked into the buffer mixture to amount to 10% concentration. Where specified PEG4000 (Sigma-Aldrich, Gillingham) was spiked into the buffer mixture to amount to 10% concentration. All assays were conducted at 37 degrees Celsius.

### 2.4. Measurement of Turbidity at OD600 

Turbidity was measured using Polar Star Optima by BMG laboratories (Offenburg, Germany) at an absorbance of 600 nm. The assay was conducted at 37 degrees Celsius.

### 2.5. RNA and TDP-43 LCD Microscopy

Total yeast RNA (Sigma, Gillingham, UK; Cat N° 10109223001) was prepared in 2-(N-morpholino)ethanesulfonic acid (MES) buffer (20 mM, pH 7.0) or phosphate buffer (50 mM, pH 7.5). For fluorescence microscopy, we spiked the phase-separating mixture with TDP-43 labeled with Alexa Fluor™ 488 at 1:100 ratio. We incubated the samples for 5 min (unless otherwise stated) in incubation chambers (grace biolabs, 666,106 JTR12R-A2-1.0) sealed with 1.5# coverslips coated with PEG silane. Samples were visualized by fluorescence microscopy (Nikon A1R Eclipse Ti, Tokyo, Japan) using a 488 laser and Apo 60X objective. Images were taken in approximately the middle of the incubation chamber to avoid wetting effects on the coverslips.

### 2.6. Atomic Force Microscopy (AFM)

For samples analyzed in air, aliquots of TDP-43 LCD time points (100 µL) were placed on a clean, freshly cleaved grade V-1 mica (AGG250-1: 75 × 25 × 0.1 mm, Laborimpex). After 10 min of adsorption, the solvent was removed by filter paper successively in order to remove salts. Samples were dried overnight. AFM images were acquired in PeakForce QNM mode on MultiMode 8 (Bruker Santa Barbara, CA, USA) with ScanAsyst-air tips. AFM images were analyzed using the NanoScope Analysis 1.9. For samples analyzed in liquid, 100 µL of TDP-43 LCD in phosphate buffer and/or MES buffer were placed on clean, freshly cleaved mica sheets (AGG250-1: 75 × 25 × 0.1 mm, Laborimpex). After different indicated time points of adsorption, AFM images were acquired in PFQNM mode on multimode VIII (Bruker-Santa Barbara) with ScanAsyst-air tips. AFM images were analyzed using the NanoScope Analysis 1.9.

### 2.7. FRAP

Fluorescence recovery after photobleaching (FRAP) assays were performed in a black, low binding 96-well plate with clear bottom using the Leica dfc7000 microscope with Leica WF FRAP system. Protein was labeled with DyLight 488 ester (Thermo Fisher Scientific Rockford, IL, USA). LLPS was induced, and the sample was incubated for 30 min. Stationary droplets were bleached (100% intensity, 500 ms bleach, 5 bleaches with 50 ms between each bleach), and observed every second. To study single bleached droplets, the fluorescence of a single droplet was quantified with ImageJ. To study droplet recovery, data were normalized using following formula:$$Recovery=\;\frac{Fluo-Fluo_{bleach}}{Fluo_{pre-bleach}-Fluo_{bleach}}$$

Recovery half-time values and plateau values were obtained by fitting the data with Prism one-phase association.

### 2.8. Dynamic Light Scattering

Dynamic light scattering (DLS), also known as photon correlation spectroscopy or quasi-elastic light scattering, measures the fluctuation of intensity of scattered light with time. Using a DynaPro DLS plate reader III instrument (Wyatt, Santa Barbara, CA, USA) equipped with an 830 nm laser source, we determined the hydrodynamic radius of the particles. One hundred microliters of each sample (containing 20 µM of TDP-43 LCD) was placed into a flat-bottom 96-well microclear plate (Greiner, Frickenhausen, Germany). Ten image acquisitions of particles were taken at approximately 150x of data per replicate (replicate *n* = 3). Data were acquired, and Wyatt Dynamics software 7.8.1.3 was used to calculate the average radius. 

### 2.9. Transmission Electron Microscopy (TEM)

Samples of proteins at different time points were placed onto formvar- and carbon-coated copper grids (Agar Scientific Ltd., Stansted, England) and incubated for 5 min. Excess protein was removed using filter paper from the edge of the grid. The grids were washed 3 times with filtered distilled water, and blotted dry on filter paper. Next, negative staining was conducted with 2% filtered uranyl acetate for 2 min. Excess dye was blotted on filter paper, washing was conducted again 3 times, and samples were left to air dry for 10 min. Micrographs were obtained using a 120 kV JEM-1400 transmission electron microscope (JEOL, Tokyo, Japan) at accelerating voltage 80 keV.

### 2.10. Western Blotting

For Western blotting, samples were boiled at 95 °C for 10 min. Each sample was loaded onto 10-slot 4–20% precast polyacrylamide gels (#4568094, Bio-Rad). The gels were run at 100 V for approximately 1 h. Next, the gels were stained with Coomassie^®^ R-250 (Bio-Rad, Watford, UK) shaker at 4 °C overnight. Next, the gel was washed with deionized water before being de-stained with de-staining solution (containing 10% ethanol and 7.5% acetic acid).

Western Blots were imaged with the ImageQuant LAS 4000 Biomolecular Imager (GE Healthcare, Chicago, IL, USA). Protein bands were quantified with ImageQuant™ TL version 7.0 software (Cytiva, Buckinghamshire, UK).

### 2.11. Sedimentation Analysis

While complex mixtures of aggregated protein can be imaged by a variety of imaging and plate reader techniques, the separation of insoluble species can be challenging. To gain insight into how much seeded assembly contributes in comparison to spontaneous assembly, we used sedimentation techniques. Insoluble fibrils and larger aggregates pellet quickly around 14,000–16,000× *g*, and smaller species such as oligomers or shorter fibril fragments sediment more slowly under high-speed centrifugation 20,000× *g*. Samples at end-point (15 h) were placed in a new Eppendorf and spun at 20,000× *g* for 1 h at 4 °C to ensure that both fibrils and oligomers sedimented. The average soluble concentration of 3 readings was then read on Thermo Scientific Nanodrop2000 using the molar extinction of TDP-43 LCD.

## 3. Results

### 3.1. TDP-43 LCD Undergoes Buffer-Dependent Liquid–Liquid Phase Separation and Aggregation

LCDs of multiple proteins have been shown to undergo spontaneous LLPS in the test tube, and this process is affected by conditions such as pH, salt concentration, molecular crowding and temperature [24,36]. We compared the behavior of the LCD of TDP-43 in phosphate buffer (pH 7.5) versus MES buffer (pH 5.5). The MES buffer, hereafter also referred to as LLPS- buffer, is the one in which the protein is solubilized during the last stage of purification. As shown in Figure 1A, TDP-43 LCD undergoes LLPS in the phosphate buffer, hereafter referred to as LLPS+ buffer in comparison to the MES buffer (LLPS- condition). Using fluorescence recovery after photobleaching (FRAP), we observed a 65% recovery rate after photobleaching in the LLPS+ buffer (Figure 1A). As no droplets formed in the LLPS- condition, FRAP could not be conducted for this condition. In LLPS+, the size of particles starts to increase immediately with the formation of droplets, which get bigger by the process of Oswald ripening (Figure 1B). At 2 h, they reach a maximum size of 1.2 μm. In LLPS- buffer, the formation of (non-droplet-like) particles progresses much slower and they start to increase to an average size of 0.2 μm only after 3–4 h (Figure 1E). Next, we compared TDP-43 LCD turbidity, referred to here as optical density at 600 nm (OD600), and ThT fluorescence (Figure 1C,F). We observed that the OD600 values are higher in LLPS+ buffer at 0 h compared to 4 h, i.e., turbidity appears to follow the kinetics of LLPS. In accordance, the OD600 in MES is low, regardless of the time (0 or 4 h). ThT fluorescence is similar at time 0 in both buffers, indicating the lack of aggregates at the beginning of the experiment. After 4 h, concomitant to the drop of the OD600 signal, the ThT signal increases significantly in the LLPS+ buffer, underscoring aggregation under LLPS+, but not (or much less) under LLPS- conditions. These conclusions were corroborated by atomic-force microscopy (AFM) and electron microscopy (EM) experiments at time 0 and after 6 h (Figure 1D,G). We observed signs of fibers protruding out of droplets only late in both buffers, in line with what was previously reported for TDP-43 LCD [30]. Based on fluorescence, imaging and DLS, we conclude that aggregation occurs faster in LLPS+ buffer compared with LLPS- buffer where it is slower. 

### 3.2. LLPS of TDP-43 LCD Promotes Aggregation 

Our first results suggest that aggregation of TDP-43 LCD is promoted by LLPS, as it appears more prevalent in LLPS+ buffer (phosphate buffer) in line with what was shown before [21,30]. Next, we investigated this effect in more detail. As both LLPS and aggregation are largely driven by hydrophobic interactions, we explored the effect of a broad range of buffer conditions. The obtained results were interpreted using the concept of the Hofmeister series, which ranks the influence of anions on the physical behavior of proteins, as manifested in folding, assembly and aggregation, and phase separation [30,37].

To this end, we assessed the relationship between phase separation and aggregation in 18 different buffers of different pHs, anionic components and ionic strengths (Appendix A), by recording absorbance (OD600, indicative of LLPS) and ThT fluorescence (indicative of aggregation) immediately after protein loading. An example of one of these buffer conditions (sodium citrate; 25 mM; pH 5.0) is shown in Figure 2A,B. LLPS occurs fast (within half an hour) as confirmed by the immediate cloudiness of the solution and OD600 values. Aggregation only follows later (within approximately 2 h), as indicated by ThT fluorescence. Aggregation was also confirmed by checking the bottom of the plate. The events which occur between 30 min and 2 h are crucial to understand LLPS dynamics and transition to aggregation. To resolve this issue, we compared the peak initial value of OD600 curves (OD600 0 h, a measure of LLPS propensity) and both the end-point of aggregation (a measure of total amount of aggregates formed) and the half-time of the ThT sigmoidal curves (a measure of the speed of aggregation, cf. Figure 2A,B) for the broad range of conditions (Appendix A). In the ThT read-out, we used plate shaking before each measurement to ensure that aggregates are not deposited at the bottom of the well. Detailed individual OD600 and aggregation graphs are shown in Appendix A. The propensity of the TDP-43 LCD to phase separate has a definite effect on its aggregation as the aggregation end-point increases (Figure 2C), whereas the half-time of aggregate formation decreases (Figure 2D) with increasing OD600. The aggregation end-point shows a highly non-linear behavior (hardly any aggregates form up to about PS = 0.3, but rapidly increase afterwards, with >100x difference between the two extremes), and the half-time decreases similarly in a non-linear fashion and shows only about 2x difference between the extremes. These results suggest that droplets have little effect on the kinetics of nucleation of aggregation (i.e., the time it takes nuclei to form), but they have a very positive effect on the number of successful nucleation events, i.e., nuclei formed, meaning their surfaces serve to stabilize seeds, but not to increase aggregation. 

For the underlying atomic interactions, phase separation of TDP-43 LCD has been suggested to have both hydrophobic (e.g., pi–pi) and electrostatic (e.g., charge–charge) interactions [30]. The small molecule 1,6-hexanediol has been used as an indicator of hydrophobic interactions [38,39]. In our experiments, 1,6-hexanediol decreases both TDP-43 LCD phase separation and subsequent seeded aggregation (Appendix A), suggesting that hydrophobic interactions play a role in both LLPS and aggregation. 

### 3.3. TDP-43 LCD Aggregates Effectively Seed Further Aggregation

As previously described, aggregated TDP-43 can seed its own aggregation [40]. We also observed that LCD aggregation can be accelerated by adding aged sonicated TDP-43 LCD seeds to the TDP-43 LCD solution at a 1:20 seed:protein ratio. Seeding dramatically increases in the LLPS- environment (MES buffer, Figure 3A) in comparison with LLPS+ conditions (phosphate buffer, Figure 3B). In LLPS- conditions, the half-life of aggregation decreases whereas its end-point (total aggregate formed) increases. This implies that despite droplets leading to protruding fibers in TDP-43 LCD alone, once additional seeds are added, there is an inefficiency in further detecting a seeding phenomenon. As droplets promote the effectivity of primary nucleation, but much less secondary nucleation, by seeds already formed (Figure 3), this parallels the effect that droplets apparently stabilize nuclei formed but not their kinetics of formation (Figure 2). The effect of a crowder polyethylene-glycol (PEG) which promotes seeding (Appendix A) is compatible with this model. PEG 10% is known to increase LLPS of TDP-43 and other proteins [28,30]. However, we observed that PEG10% can also influence seeding aggregation kinetics. When seeding is occurring (Appendix A), the effect of crowding minimizes the change observed with addition of seeds.

As the ThT values were much higher in MES buffer during TDP-43 LCD seeding, we wanted to determine what proportion of TDP-43 is in an aggregated state as it may not be clear whether elevated ThT fluorescence corresponds to more TDP-43 aggregation or a different strain of TDP-43 fibril binding ThT. Centrifugation protocols have been used to separate soluble from insoluble fractions [41,42]. Insoluble fibrils and larger aggregates pellet quickly around 14,000–16,000× *g*, and smaller species such as oligomers or shorter fibril fragments sediment more slowly even under high-speed centrifugation. The samples containing protein and seed at end-point (15 h) were centrifuged and the soluble fractions compared. The soluble fraction of the LLPS- condition is 13% less soluble than the LLPS+ buffer condition confirming that indeed there is more aggregated TDP-43 LCD in the non-LLPS condition, which corresponds to the higher ThT signal (Figure 3 and Appendix A).

### 3.4. Phase Separation Lowers the Kinetics of Seeded Aggregation

Based on our observation that LLPS conditions influence early seeding dynamics, we predict that droplets behave as “seeds” themselves as previously described for hnRNPA1 [43]. An open question is how further seeding influences this state. In order to test this, we first selected seven regions of TDP-43 LCD with a potential function as APRs [44] (peptides 1 to 6, as well as the recently described LARKS (low-complexity aromatic-rich kinked segments) region, cf. Table 1; Figure 4A). These LARKS stack into reversible (Velcro-like) associations and have been proposed to form distorted β-sheets stabilized by inter-strand hydrogen bonds or aromatic stacking interactions [45]. In addition, LARKS are hypothesized to play a role in the reversible process of TDP-43 LCD as well as pathogenic aggregation by bringing adjacent amyloid-forming segments together [44]. To assess their APR characteristics, we first assayed these peptides for aggregation. In general, all peptides show some tendency to undergo aggregation, but the morphology of their aggregates (either with or without sonication, Figure 4B) show some variation ranging from shard-like structures (peptide 1), to more amorphous structures (peptide 5), to more fibrous structures (peptide 3). Next, we addressed seeding of LLPS and aggregation by preformed peptide amorphous aggregates/fibers, by mixing pre-aggregated, sonicated peptide fibrils into TDP-43 LCD solution, in both MES (LLPS-) and phosphate (LLPS+) buffer at 1:20 molar peptide: protein ratios. While three out of seven peptides (peptides 4, 5 and LARK) do not affect aggregation in the LLPS- buffer, the remaining peptides (peptides 1, 2, 3 and 6) do seed aggregation (Table 1; Figure 4C).

The peptides do not have an effect on LLPS itself. Changes in LLPS behavior using seeded peptides are negligible as can be seen in Appendix A, but they alter aggregation as can be seen with ThT (Figure 4C). 

The peptides show seeding properties in different ways. Using ThT data, peptides 1 and 3, in the presence of droplets, promote an earlier aggregation onset, but result in the same final aggregation level. Peptide 6, the longest peptide that includes the sequence of peptide 3, aggregates more in non-droplet (LLPS-) conditions, i.e., while it aggregated faster when TDP-43 LCD is in the form of droplets, it aggregates to a higher extent without droplets being present. These results show that specific nucleating region(s) exist in TDP-43 LCD, which are within peptide 6 that roughly covers peptides 1 and 3. When comparing peptides, it should be noted that peptide 6 is of the longest length. We can compare this to Figure 3 when TDP-43 LCD seeds were added which comprise of the longest seeder 147 residues long. This has a very dramatic effect in the LLPS- buffer in comparison to LLPS+ buffer. We suggest two nucleation paths for TDP-43 LCD aggregation. In one, droplets formed by LLPS are the primary site of nucleation, and peptide (seeds) can cooperate with droplets. In the absence of LLPS, peptide seeding is slower, but results in the same or even higher final level of aggregation. Interestingly, peptides with particularly large effects (peptides 3 and 6) encompass the helical region LCD (320-330) [21], which is involved in LLPS. Addition of the molecular crowder PEG enhances TDP-43 LCD LLPS, coinciding with an increase in subsequent aggregation (Appendix A), which suggests that seeds interact with droplets in a reaction that reduces molecular volume. Seeding cannot be distinguished so easily in LLPS+ conditions promoted by molecular crowders (Appendix A). This feature of seeding aggregation does not depend on prior LLPS, i.e., aggregation with and without the formation of droplets is similarly promoted by the addition of PEG.

### 3.5. RNA Influences the Liquid–Liquid Phase Separation of TDP-43 LCD 

Recent work implies an intricate interplay between RNA and proteins in the LLPS of RNA-binding proteins [31,46]. As a consequence, we checked how RNA influences the LLPS and seeding of TDP-43 LCD. We investigated the effect of total yeast RNA on the formation of TDP-43 LCD droplets in phosphate buffer (LLPS+ condition) using DIC microscopy, to determine the possible influence of RNA binding on the LLPS and/or aggregation of TDP-43 LCD (Figure 5A). We observed that the number of TDP-43 LCD droplets increases significantly with increasing RNA concentration, up to 400 ng/µL, where we observed twice as many droplets (Figure 5B), while their size (surface area) becomes significantly smaller (Figure 5C,D), i.e., RNA appears to have an emulsifying effect on TDP-43 LCD droplets. At higher RNA concentrations, both trends seemed to reverse, suggesting a “reentrant” behavior of the RNA effect [47,48]. The underlying process is reversible as the addition of RNase reverses both the average area and the number of droplets back to the initial conditions (no RNA) (Figure 5D,E). Overall, this effect is intriguing as the LCD of TDP-43 harbors no proven RNA-binding motif or domain. Whereas LLPS (and aggregation) of TDP-43 LCD relies on both hydrophobic (as demonstrated by the effect of 1.6-hexanediol, Appendix A) and electrostatic (as evidenced by charge screening observed in Appendix A and in line with ref. [30]) influences, the observed RNA effects might be due to the interaction of the positively charged LCD (pI = 10.78) and negatively charged RNA. Such “reentrant” behavior is a sign of the dominance of electrostatic screening, especially at a high RNA stoichiometry, which cannot be accounted for by a well-defined RNA binding site on the protein [48]. 

### 3.6. RNA Initiates More Aggregation of TDP-43 LCD via LLPS 

As we have observed that RNA has a strong effect on the LLPS of TDP-43 LCD, and LLPS provides a mechanism of aggregation, we tested whether RNA exerts an effect on the aggregation of TDP-43 LCD induced through LLPS. First, we observed that the addition of total yeast RNA to TDP-43 LCD results in higher turbidity (OD600), compatible with the formation of more (smaller) droplets already observed before (Figure 5A–C and Figure 6A). In agreement with our previous observations on the positive link between LLPS and aggregation, the formation of more droplets leads to more aggregation, as seen by ThS fluorescence (Figure 6B, ThS was used because of the ThT binding to RNA). By electron microscopy (EM), it is also apparent that not only the amount of aggregates, but also their morphology is very different (Figure 6C). In the absence of RNA, a few very long fibers are seen, suggestive of infrequent nucleation events. In the presence of RNA, large clumps of aggregates suggested frequent nucleation that dominates aggregation. In agreement with these observations, as shown in Figure 6D,E, the amount of TDP-43 LCD is higher in the supernatant before the incubation (0 h) of TDP-43 LCD with RNA under LLPS- conditions (MES buffer), while the relative amount is shifting more to the pellet at a later time point, 4 h after combining TDP-43 LCD with RNA. In LLPS+ conditions (phosphate buffer), the LCD-TDP-43 is higher in the pellet and even higher when combined with RNA, and this is independent of the time. 

## 4. Discussion

In this study, we observed that the formation of TDP-43 LCD droplets by LLPS enhances the formation of aggregates. This was demonstrated using three different approaches: by altering the buffer conditions, by adding seeds and by combining the protein with RNA.

Placed in favorable conditions, droplets formed by LLPS of a broad range of proteins are initially highly dynamic, liquid-like [11], but almost inevitably undergo maturation into more solid-like states. Whether these solids are labile fibrils, hydrogels or pathological amyloids depends on the actual protein and conditions [30]. These transition(s) have been observed for many proteins, e.g., FUS, tau and TDP-43, yet details of the underlying mechanism(s) remain unclear. Even the most basic question, whether LLPS is mandatory for subsequent aggregation or whether aggregation may occur without the prior formation of liquid droplets, depends a lot on the protein studied and the experimental approaches used. A specific caveat to the underlying mechanisms is that none of the molecular species appearing in seeding, LLPS and aggregation correspond to—and can be described as—equilibrium states. As a consequence, the underlying transient, dynamic phenomena are inherently sensitive to experimental conditions and stochastic effects. In other words, studying them under a broad range of conditions is required to draw definite conclusions.

One of the most intriguing proteins to study for the link between LLPS, aggregation and disease is TDP-43. This protein is found in aggregates in ALS and FTLD and readily undergoes LLPS, which very often appears to be linked to aggregation [30]. Interestingly, a deep mutagenesis screen of more than 50,000 mutants in the LCD of TDP-43 discovered that changes in hydrophobicity and aggregation potential are linked to toxicity in yeast [49]. 

We used a buffer screen to test the propensity of TDP-43 LCD in multiple conditions to dissect the relationship between LLPS and aggregation. We observed that every condition favoring LLPS also significantly promoted aggregation. In general, this can be interpreted in terms of two types of mechanism of aggregate formation, either by “homogeneous” (within droplets) or “heterogeneous” (on the surface of droplets) nucleation of droplets.

By conducting seeding experiments using synthetic peptides and the LCD of TDP-43 itself, we obtained mechanistic details on the role of TDP-43 LCD seeding in the context of LLPS, which, to our knowledge, has not been previously addressed. While all the peptides seeded aggregation in an LLPS setting, peptide seeding (peptides 1, 3 and 6) was more dramatic in non-LLPS conditions. This implies that not only do these regions probably function as APRs and promote the amyloidogenic behavior of this protein, but they depend on the protein buffer situation. TDP-43 seeding was investigated previously both in cells and in vitro, as demonstrated by the plethora of evidence comprising: human to animal propagation, human to cell propagation, cell to cell propagation, cell seeding with recombinant protein, in vitro seeding with recombinant protein, fibril formation [50,51,52,53]. In this study, sequence lengths and actual sequences were somewhat different. Three of our peptide sequences align with the six steric zippers discovered previously [44], namely ^328^AALQSS^333^, ^321^AMMAAA^326^ and ^333^SWGMMGMLASQ^343^ (the latter (peptide 4) with the addition of one glutamine, is almost identical). ^321^AMMAAA^326^ and ^333^SWGMMGMLASQ^343^ belong to the pathogenic/mutational aggregation core. Potential mutation sites as shown in Figure 5A are indicated with an asterisk. Guenther et al. confirmed with mutagenesis that all six short segments are important for TDP-43 C terminal (CTF) aggregation [44]. Four other sequences in their study were labeled as LARKS, labile amyloid-like interaction sites characteristic of protein hydrogels and parts of proteins responsible for SG dynamics. Such LARKS have also been identified in FUS [54] and hnRNPA1 [45]. Although the function of these LARKS is not yet clear, we investigated whether the LARKS regions behaved differently compared with our other peptides, because LARKS (312-317) was very similar to an APR found in our initial screen (313-319), with only one residue less: ^312^NFGAFSI^317^. While the Eisenberg group showed using EM that LARKS can form microcrystals, we observed fibers using EM. Mutations in this region are said to disrupt phase separation but not aggregation [44]. We observed that the addition of the LARKS peptide in a non-LLPS setting did not act as a seeder, while we observed seeding in an LLPS+ setting.

We interpret these results in the context of particular structural features of the peptides assessed by the Zipper DB algorithm [55,56]. Using this algorithm, peptide 1 contains a predicted zipper, peptide 3 contains a LARK segment which induces kinks, but capped zippers still aggregate and zippers can fold back on themselves to form mating sheets. Peptide 6, the longest region, contains a LARK segment which is capped by zippers, which are more likely to form long uninterrupted beta sheets. In peptide 4, which includes residues of peptide 3, there might not be enough steric-zipper content to overcome the break in beta-sheet structure. Based on the Zipper DB algorithm, it has previously been suggested that mutations in regions of steric zippers influence the lag phase of aggregation [55,57]. Although the deletion of hexapeptide steric zippers in hnRNPA1, another ALS-related protein, eliminated fibrillization, it did not abrogate phase separation [43]. Therefore, we suggest that classical seeding studies involving proteins which can undergo LLPS should be conducted in both settings, under LLPS+ and non-LLPS conditions, as potential seeding events could be missed. The presence of droplets influences seeding dynamics.

In a disease-related context, it is not yet clear whether RNA assemblies, such as SGs, besides their regular function, are indeed harmful intermediates of aggregates. This conjecture was favored because of the link of neuronal vulnerability to stress, which is implicated in aging. However, stress granules are not ALS-specific, and the absence of many SG proteins in inclusions has questioned this hypothesis. Many studies have been conducted to establish the number and composition of proteins in SGs, although their results heavily depended on the methods used to promote and isolate SGs [58,59,60,61]. In addition, there is evidence for an initial “protective” effect of these granules [33,62]. In contrast, others reported that RNA binding prevents TDP-43 phase separation [31,33], suggesting that droplets deplete the amount of TDP-43 available for its RNA-related functions. 

To further address this issue, we studied the effects of RNA on the TDP-43 LCD phase separation and aggregation [31,63]. RNA appears to be important in a physiological context, by controlling the balance of physiological and pathological phase separation in various locations of the cell [31]. We discovered that the addition of RNA to TDP-43 LCD increases its LLPS. This leads to further aggregation, probably due to RNA contributing to seeding TDP-43 LCD aggregation. This demonstrates again that more LLPS leads to more aggregation. While we found that RNA influenced TDP-43 LCD in a “reentrant” fashion, i.e., with a maximal effect at an intermediate concentration, it appears to have both hydrophobic and electrostatic components. For full-length TDP-43, it has been shown that physiological oligomerization of TDP-43 is mediated through its N-terminal domain [64], which forms functional and dynamic oligomers [65]. In our study, the TDP-43 LCD lacks this N-terminal region of the protein, the presence of which could modulate the interaction of LCD monomers, perhaps even preventing it from aggregation. 

Furthermore, it should also be noted that TDP-43 LCD contains an arginine- glycine, -glycine (RGG) motif previously found to interact with RNA [66], and using mass spectrometry it was found that RNA could stick to the phenylalanine-glycine (FG)-rich LCD 275-293 (FGGNPGGFGNQGGFGNSR) [67], i.e., the observed effect could also be indicative of a direct, specific interaction between the LCD and RNA.

The regions which seed aggregation used in this study could be used to generate aggregation-based models in early embryonic cells, such as induced pluripotent stem cell (iPSC)-derived motor neurons, to generate physiologically relevant models of pathology, as these models lack end-stage phenotypes attributed to their premature nature. Furthermore, peptide-specific aggregation blockers could be potentially generated as methods of therapy not just to influence aggregation but LLPS as well. However, as demonstrated in our paper, the method with which seeding for an LLPS prone protein is studied is critical. 

The results obtained in this study contribute to our understanding of how LLPS can accelerate aggregation and why the LCD of TDP-43 is so sensitive to environmental changes. Based on our findings, we conclude that droplets formed by LLPS behave as aggregators for TDP-43, and RNA can further enhance this effect. We conclude that the level of aggregate seeding is weakened when droplets are present, despite droplets allowing seeding to occur faster.

## Figures and Tables

**Figure 1 biomolecules-11-00548-f001:**
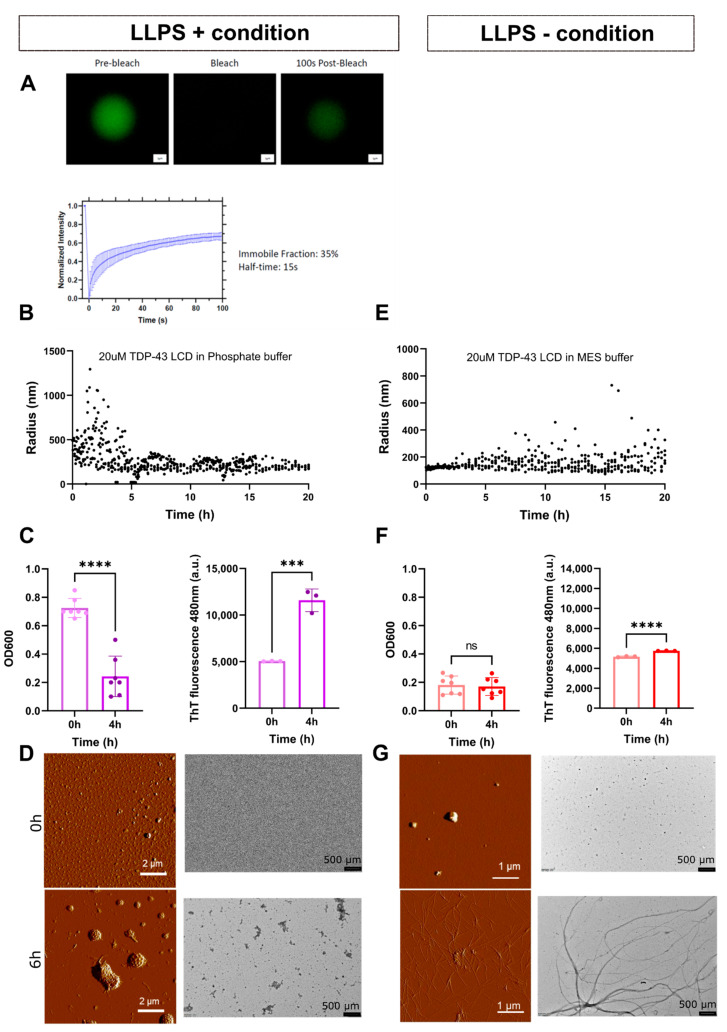
Characterization of the TDP-43 LCD behavior in phosphate buffer (LLPS+) and in MES buffer (LLPS-). (**A**) Fluorescence recovery after photobleaching (FRAP) of labeled TDP-43 LCD (20 µM) showing droplet recovery in the LLPS+ condition (50 mM phosphate buffer, pH 7.5) for three individual experiments. FRAP was not possible for the LLPS- condition (20 mM MES buffer, pH 5.5) as there were no droplets present. Examples of the droplets at different time points are shown on top of the panel. Scale bar = 1 µm. (**B**) Average of triplicate dynamic light scattering (DLS) data showing radius of protein within a 20 h time frame. Maximum size of droplets in LLPS+ buffer was 1.3 µm. (**C**) Turbidity (OD600) and fluorescence (ThT) values at time 0 h and 4 h. Statistical differences determined with an unpaired Student’s *t*-test with *n* = 3 for ThT graphs to look at aggregation and *n* = 7 for OD600 to look at LLPS absorbance at time 0. Data are presented as mean  ±  SD, **** *p*  <  0.0001, *** *p* < 0.001. (**D**) Atomic force microscopy (AFM) and electron microscopy (EM) of TDP-43 LCD at 0 h and 6 h. (**E**) DLS data of TDP-43 LCD in MES buffer without LLPS. (**F**) Turbidity and absorbance of TDP-43 LCD in MES buffer at 0 h and at 4 h. Statistical differences determined with an unpaired Student’s *t*-test; **** *p* < 0.0001. ThT *n* = 3 and OD600 *n* = 7, mean ± SD is shown. (**G**) Atomic force microscopy and electron microscopy of TDP-43 LCD at 0 h and 6 h. Scale bar length is indicated on each photo.

**Figure 2 biomolecules-11-00548-f002:**
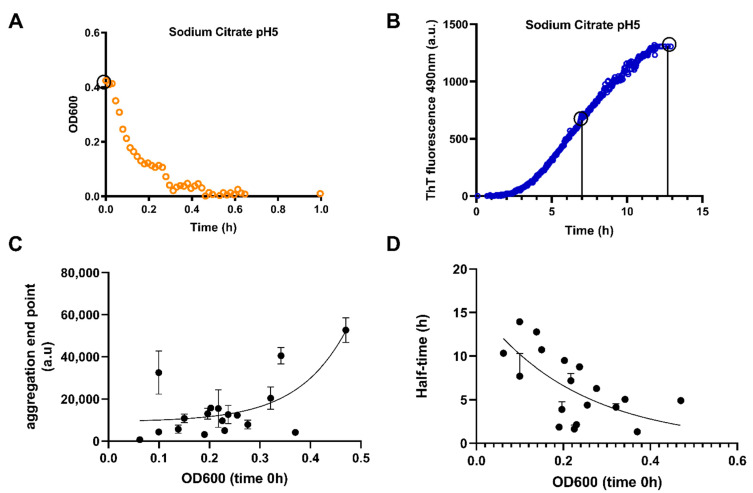
Determination of the TDP-43 LCD behavior in a variety of buffers belonging to the Hofmeister series (**A**–**D**). Representative examples of the two assays that were used to determine the behavior of TDP-43 LCD buffer screen. The measurement of the OD600 was used to determine the level of initial LLPS (**A**), and determination of ThT fluorescence at 490 nm was indicative of amyloid aggregation (**B**). The assays show the behavior of TDP-43 LCD (20 µM) in one buffer, sodium citrate 25 mM at pH 5.0. Based on similar measurements of (**A**,**B**), use of time 0 OD600, half-time of aggregation curve (using EC50 of dose–response curve fit), and aggregation end-points for each buffer belonging to the Hofmeister series were obtained as shown in Appendix A. (**C**) Data showing mean aggregation end-point versus mean OD600 at time 0 for 18 different buffers belonging to the Hofmeister series ± SD. (**D**) Half-time (EC50) of ThT fluorescence versus mean OD600 value at time 0 was plotted ± SD. Where error bars are not depicted, error was too low to plot. Black circles on graphs in panels (**A**,**B**) show how the values used for (**C**,**D**) were obtained. (**C**,**D**) fitted using non-linear sigmoidal dose–response (variable slope).

**Figure 3 biomolecules-11-00548-f003:**
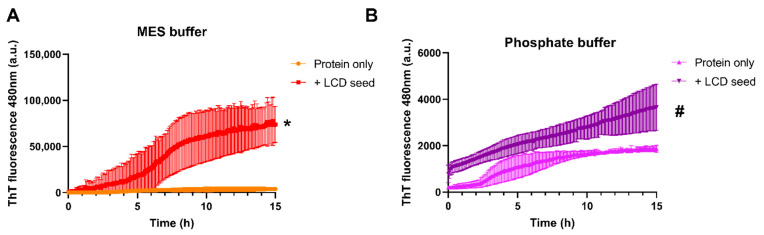
Detection of seeding in LLPS and non-LLPS promoting conditions. ThT fluorescence measurements as a function of time used to determine the aggregation speed of 20 µM TDP-43 LCD. Mean ± SD shown for *n* = 3 independent experiments. (**A**) +/− TDP-43 LCD seed added at 1:20 molar ratio to fresh TDP-43 LCD protein in MES buffer (LLPS-). (**B**) +/− TDP-43 LCD seed added at 1:20 molar ratio to fresh TDP-43 LCD protein in phosphate buffer (LLPS+). Y-axis scale for both graphs is different to allow for visualization of seeding effect. For both (**A**,**B**) conditions, the increase in ThT fluorescence was followed over time and compared with the condition without the seeds. Protein only is protein in corresponding buffers with no additional LCD seeds. Sedimentation assays conducted for * and # end-point where mixture included seeds. There was approximately 12% more soluble protein left in the LLPS+ condition (phosphate buffer) as is shown in Appendix A.

**Figure 4 biomolecules-11-00548-f004:**
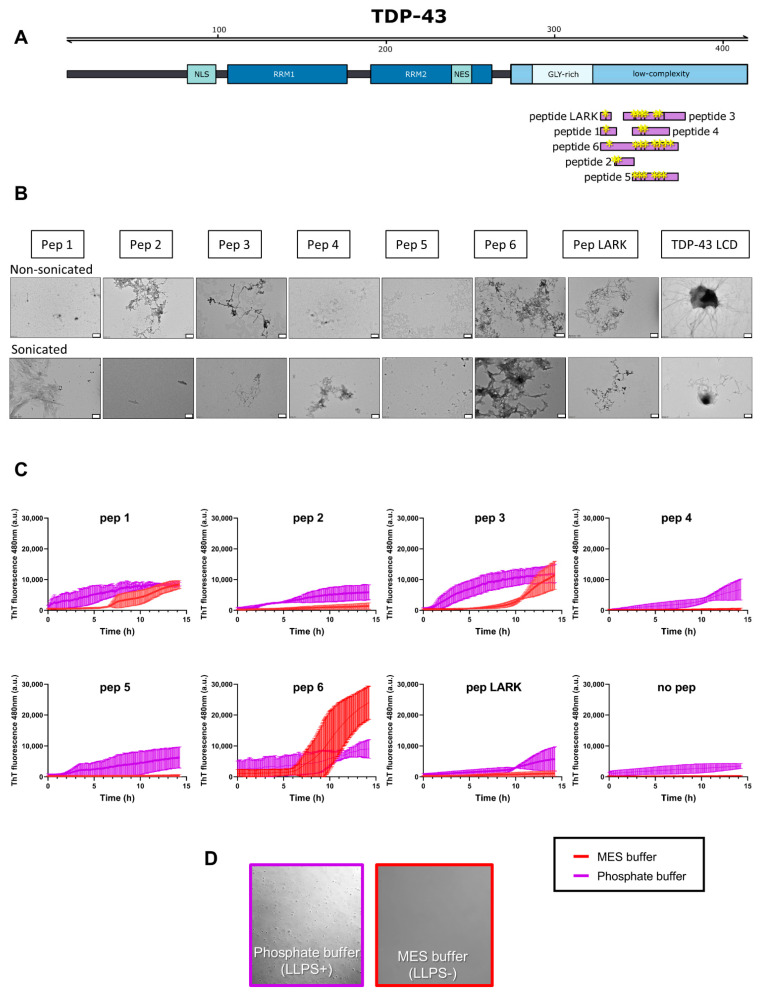
Effect of sonicated TDP-43 peptides on TDP-43 LCD aggregation. (**A**) Overview of the peptide regions within the LCD of TDP-43; yellow “*” indicates the location of potential disease-causing mutations within these peptide regions. (**B**) Visualization of the TDP-43 peptides 1–6, the LARKS peptide and TDP-43 LCD using EM at a magnification of 20,000×. Scale bar is 500 nm. Peptides were dissolved in water, filtered and left to age for 2 days. Subsequently, peptides were sonicated or not, as indicated. (**C**) ThT curves of representative sonicated peptides and TDP-43 LCD in the presence (purple) and absence of LLPS buffer (red) with TDP-43 LCD (10 µM). Mean ± SD of *n* = 3 independent experiments. (**D**) Differential interference contrast (DIC) image of 20 µM TDP-43 in LLPS+ and LLPS- buffer.

**Figure 5 biomolecules-11-00548-f005:**
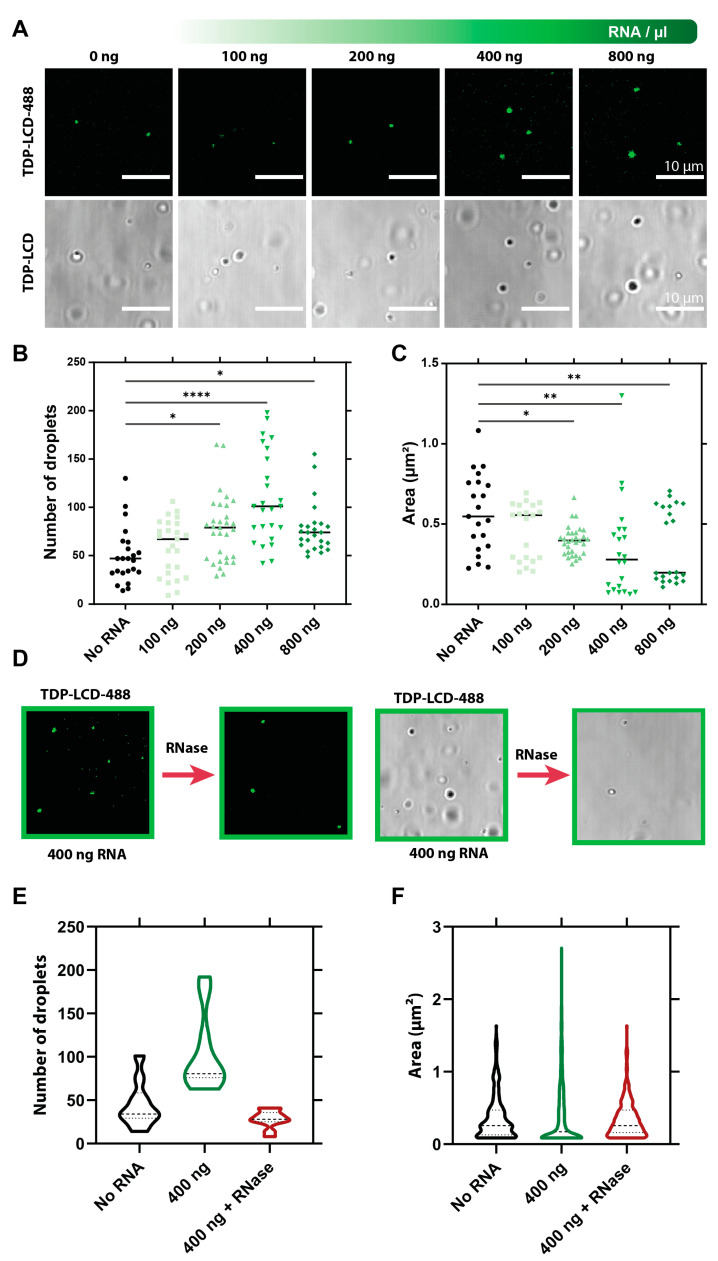
Use of RNA on TDP-43 LCD liquid–liquid phase separation. (**A**) Visualization of TDP-43 LCD droplets in LLPS+ buffer (phosphate buffer, 50 mM, pH 7.5) after a 10-min incubation with different concentrations of yeast RNA. Images taken with differential interference contrast (DIC) microscopy and fluorescence microscopy using the same fields of view, all conditions spiked in with labeled TDP-43-LCD 488 protein to visualize fluorescence. (**B**) Quantification of the number of droplets of labeled TDP-43 LCD in the absence and presence of increasing concentrations of yeast RNA in the LLPS+ buffer. (**C**) Quantification of the droplet area of labeled TDP-43 LCD. Automated analysis for (**B**,**C**) was performed for each condition. (**D**) Illustration of the effect of RNase treatment on labeled TDP-43 LCD droplet formed in the presence of yeast RNA (400 ng) after treatment with RNase (5 ng/µL). At the left side the fluorescent image is shown, while the same areas using DIC are presented on the right. (**E**) Quantification of the number of labeled TDP-43 LCD droplets with and without RNase treatment. (**F**) Determination of the labeled droplet area of TDP-43 LCD in the different conditions. Statistical evaluations were performed using one-way ANOVA with Dunnett’s multiple comparisons test. Significance is indicated compared with the control (no RNA) * *p* ≤ 0.05, ** *p* ≤ 0.01, **** *p* ≤ 0.001; *n* = 3 with an *n* = 5–10 images per replicate; mean ± SD is shown.

**Figure 6 biomolecules-11-00548-f006:**
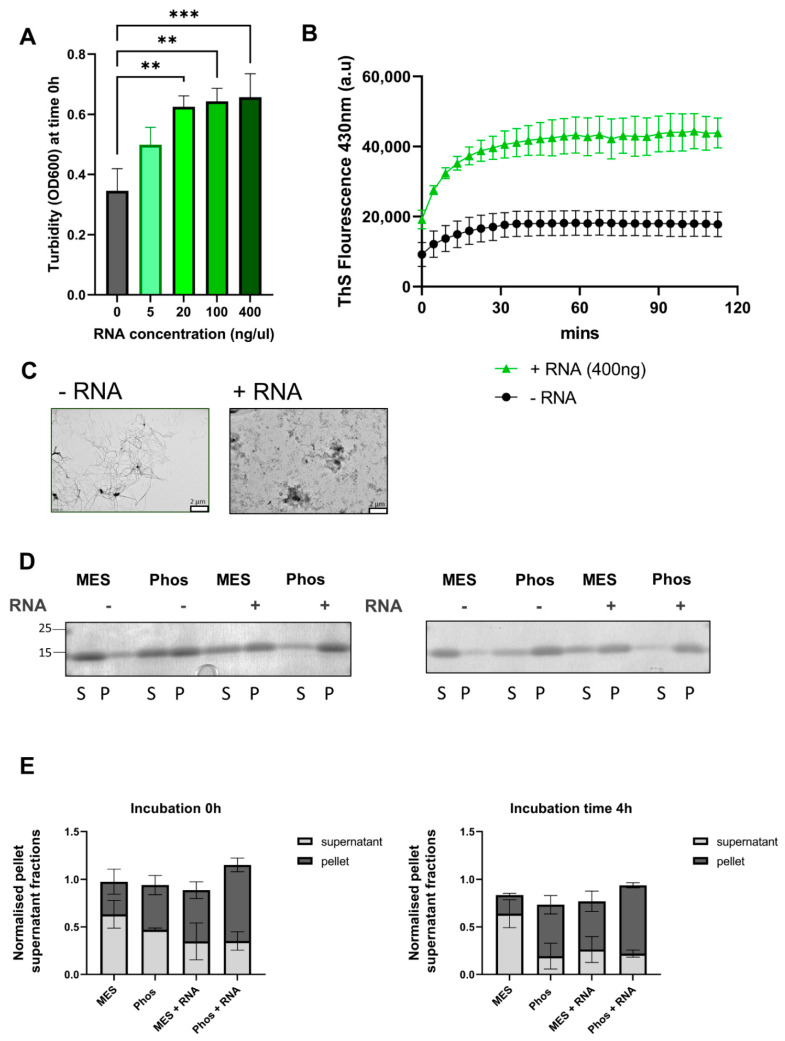
RNA increases the aggregation behavior of the TDP-43 LCD. (**A**) Addition of total yeast RNA increases the turbidity of 20 µM TDP-43 LCD in LLPS+ (phosphate buffer 50 mM) buffer; *n* = 3 mean ± SD is shown with one-way ANOVA with a Holm–Šídák multiple comparisons test used to determine significant differences. ** *p* ≤ 0.01 and *** *p* ≤ 0.001 (**B**) Addition of 400 ng/µL yeast RNA leads to a higher and faster ThS aggregation readout. (**C**) Using EM of samples of 1-week-old protein, (+/-) RNA shows two very different morphologies after addition of 400 ng of yeast RNA to TDP-43 LCD. Fibrillar morphology associated with RNA– condition. Scale bar 2 µm. (**D**) The levels of TDP-43 LCD in the supernatant (S) and pellet (P) with (+) and without RNA (-) in LLPS+ (phosphate buffer) and in LLPS- (MES buffer) was measured using Coomassie staining. The left Coomassie stain is for 0 h of incubation of protein and RNA, and the right one is after an incubation period of 4 h of protein and 400 ng/µL of RNA. (**E**) The quantification of the intensity of the bands on the Coomassie gel is shown below the blots and indicates that more TDP-43 LCD is in the pellets in the liquid–liquid phase separating conditions; *n* = 3; mean ± SD is shown.

**Table 1 biomolecules-11-00548-t001:** Characteristics and effects of the peptides selected for seeding in LLPS+ and LLPS- buffers.

Region	Peptide	Sequence	Length	Seed in LLPS-	Seed in LLPS+	Net Charge
312-327	Pep 1	**NFGAFSINP AMMAAAQ**	16	✔	✔	0
316-323	Pep 2	**FSINP AMM**	8		✔	0
329-348	Pep 3	**ALQSSWGMMGMLASQQNQSG**	20	✔	✔	0
332-342	Pep 4	**SSWGMMGMLAS**	11		✔	0
332-348	Pep 5	**SSWGMMGMLASQQNQSG**	17		✔	0
312-348	Pep 6	**NFGAFSINPAMMAAAQAALQSSWGMMGML ASQQNQSG**	37	✔	✔	0
312-317	LARK	**NFGAFSI**	7		✔	0

Sequence overlap demonstrated by sequence alignment below. ✔ indicates wherever seeding occurred.

## Data Availability

The data presented in this study are available on request from the corresponding author.

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
