# Peer review of "Liquid–Liquid Phase Separation Enhances TDP-43 LCD Aggregation but Delays Seeded Aggregation"

_biomolecules, 2021, doi:10.3390/biom11040548_

Round 1
Reviewer 1 Report
This manuscript describes the effects of three different modulators on the aggregation properties of TDP-43. This protein is interesting because associated to amyotrophic lateral schlerosis. The authors consider buffer composition (MES versus phosphate and thus pH 5.5 versus neutral), seeding and RNA. The work is potentially interesting but various improvements should be made.
a) the manuscript is not easy to read not because of problems with English but the style is heavy and dense. There are a number of places where some more detail would be appreciated, such as when the program Zipper is used. There is much confusion between zippers and LARKS. The story is complex and no prediction is shown. Could the story be simplified?
b) the story is already complex enough without unnecessary acronyms. What is the advantage of dubbing the buffers in LLPS+ and LLPS-? do we really need LARKS?
c) What is the significance of pH 5.5? It would perhaps helpful to have a comment on this point?
d) no details are given on where the RNA comes from. The authors simply mention that the use the whole yeast RNA. How was this prepared?
e) The effects of the chaotrope and cosmotrope is far too complex. It would deserve an article by itself. It is impossible to follow what is happening and the effects of the Hofmeinster series are not simple anyway. The effects of the anion and cation are of course combined and only the full screening would make sense. It also does not help that Table S1 is in Suppl. Mat. Once again, there is no need to introduce an acronym to indicate cosmotropes and chaotropes. The conclusions from this analysis (that looks completely arbitrary) are not sufficiently analysed (as said above, they would require the whole paper). I would leave this part of the story out.
f) The effect of crowding: as it is presented is hardly comprehensible. It is not mentioned in the M&M. The reader meets crowding at the beginning of the results. The relevant figures are in the Suppl. Mat. The effects should be properly commented in an indipendent paragraph dedicated to this aspect and not intertwined with the other effects.
g) The figures are far too dense.
h) The authors repeatedly refer to a cartoon of the domain architecture of TDP-43 as the structure of the protein. That is not the structure!
Minor:
The impression is that the literature is out of sync. In any case some of the citations do not appear in brackets.
Reviewer 2 Report
Pakravan et al. applied ThT, turbidity assay (O.D. 600 nm), and microscopic techniques on the low complexity domain (LCD) of TDP-43 under various buffer conditions, or in the presence of different peptides or RNAs. They concluded that: (1) TDP-43 LCD aggregates faster under the LLPS condition. (2) LLPS delays seeded aggregation.
Although this manuscript is well-written and assembles an extensive set of data, the major concern about pH prevents my recommendation for its publication.
The effect of pH is ignored through-out this manuscript. In the so-called LLPS+ buffer, the pH is 7.5; whereas in the LLPS- buffer, the pH is 5.5. The LCD has 3 negatively and 6 positively charged residues, making the total charge of +3. In the Fawzi group’s paper (2016 Structure, reference ), they have demonstrated that the NaCl promotes the LLPS because of screening the repulsive electrostatic interaction. Lim’s group (2016 PLoS Biology) conducted their experiments at pH 4 and no LLPS was observed in their studies. And as commented by Mompean et al (2016 PLoS Biology DOI:10.1371/journal.pbio.1002447 ), the electrostatic repulsion governs LCD’s aggregation. The pH changes, the net charge of the sample changes. In the instance of the LCD, the lower the pH, the net positive charge is stronger, the less tendency to LLPS, and vise versa. Therefore, their interpretation from Fig.1 to 5 is possibly caused by pH, instead of the buffers. The comparison of different salt but ignoring the pH is meaningless (Table S1). Without resolving this issue, the following experiments and the conclusion of this article are not convincing at current stage.
Other
The temperature should be stated (for example the ThT assay), as the LLPS of this LCD is also temperature-dependent.
The results of Figure 5. Is it possible that the effect is purely length-dependent? the should be ruled out (e.g. scrambling the synthetic peptides’ sequence)
Page 2 Line 74: “LCD 23”: reference formatting error?
Page 2 Line 75: a “.” is missing in “……a transient helix within the LCD [21,24] The LCD…..”
Page 9 Line 314: reference formatting error.
Round 2
Reviewer 2 Report
The authors did a substantial revision on the manuscript, in particular, the removal of the results of kosmotropic/chaotropic buffers makes the theme more clear and less ambiguous.